# Analysis of Electrodermal Signal Features as Indicators of Cognitive and Emotional Reactions—Comparison of the Effectiveness of Selected Statistical Measures

**DOI:** 10.3390/s25113300

**Published:** 2025-05-24

**Authors:** Marcin Jukiewicz, Joanna Marcinkowska

**Affiliations:** Faculty of Psychology and Cognitive Sciences, Adam Mickiewicz University, 61-712 Poznan, Poland

**Keywords:** electrodermal activity, cognitive reaction, emotional reaction, signal processing, signal analysis, psychophysiology

## Abstract

This study investigates which statistical measures of electrodermal activity (EDA) signal features most effectively differentiate between responses to stimuli and resting states in participants performing tasks with varying cognitive and emotional reactions. The study involved 30 healthy participants. Collected EDA data were statistically analyzed, comparing the effectiveness of twelve statistical signal measures in detecting stimulus-induced changes. The aim of this study is to answer the following research question: Which statistical features of the electrodermal activity signal most effectively indicate changes induced by cognitive and emotional reactions, and are there such significant similarities (high correlations) among these features that some of them can be considered redundant? The results indicated that amplitude-related measures—mean, median, maximum, and minimum—were most effective. It was also found that some signal features were highly correlated, suggesting the possibility of simplifying the analysis by choosing just one measure from each correlated pair. The results indicate that stronger emotional stimuli lead to more pronounced changes in EDA than stimuli with a low emotional load. These findings may contribute to the standardization of EDA analysis in future research on cognitive and emotional reaction engagement.

## 1. Introduction

Electrodermal activity (EDA), also known as changes in the electrical properties of the skin, is associated with factors such as sweat gland activity. The term EDA was described in [1] as encompassing all electrical phenomena of the skin. The EDA signal consists of a tonic component representing slow-changing baseline skin conductance and a phasic component, which includes short-term electrodermal responses triggered by stimuli. The course of an electrodermal response is illustrated in Figure 1. EDA is widely used in psychophysiology due to the ease of eliciting reactions and the low cost of equipment [2].

EDA signals play a significant role in studying cognitive and emotional reactions. EDA changes are linked to sympathetic nervous system activity, which increases in response to mental effort and cognitive demands [3]. Studies indicate that tasks with higher cognitive and emotional reaction lead to increased EDA [3]. An increase in EDA accompanies responses to stress, emotions, and cognitive reactions [4]. Studies [5] have shown that EDA levels rise under increased cognitive and emotional reactions, such as performing complex tasks. EDA plays a significant role in objectively measuring cognitive and emotional reactions and is often used alongside participants’ subjective reports. It is worth noting that EDA appears to be more effective for measuring cognitive engagement than affective engagement [6].

**Figure 1 sensors-25-03300-f001:**
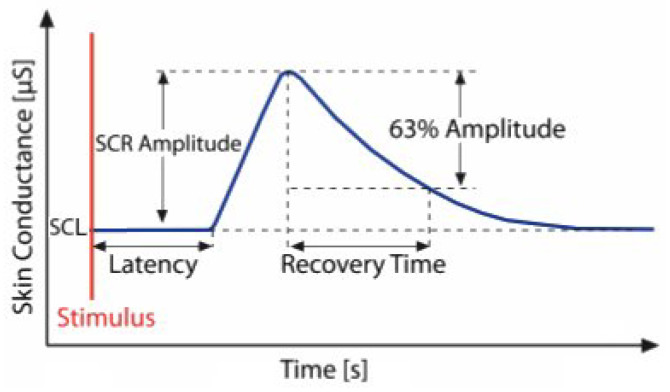
The figure illustrates the course of a single electrodermal activity event and its morphological features: baseline skin conductance level (SCL), skin conductance responses (SCR), amplitude, latency, and recovery time [7].

The study [8] compared reported emotions with physiological responses to realistic stimuli. This research highlighted (among other findings) the absence of a standardized method for analyzing EDA signals, as different authors used various signal features (mean, variance, standard deviation, etc.) to determine the occurrence of a stimulus-response. The present study aims to address this gap by identifying which statistical measures of EDA most effectively differentiate responses to stimuli from resting states. In other words, the study seeks to determine which EDA signal characteristics best indicate stimulus-induced changes, potentially contributing to the standardization of EDA analysis in future cognitive and emotional reaction research.

To achieve this objective, an experiment was conducted measuring EDA during tasks of varying cognitive and emotional reaction levels. Following data acquisition, the study’s primary focus was on advanced signal processing techniques and statistical analyses to identify the most informative signal features for reliably detecting stimulus-driven changes in EDA. It is important to emphasize that this research primarily addresses methodological issues related to the effective extraction, selection, and interpretation of statistical measures derived from EDA signals. Therefore, rather than proposing or testing novel sensor technologies or measurement hardware, this study aims to contribute to the standardization and optimization of analytical practices already established within psychophysiological research. With this detailed methodology applied, comprehensive statistical analysis results, and their interpretation within the context of existing research on signal processing and cognitive and emotional reactions, are presented in the following sections.

At this point, it is worth emphasizing and summarizing the main research question of this article, which addresses the absence of a standardized methodology for analyzing EDA signals in studies of cognitive and emotional reactions—impeding comparability across studies—by investigating which statistical features of the EDA signal most effectively reflect changes induced by these reactions, examining inter-feature similarities to identify redundant measures, and testing the central hypothesis that amplitude-based metrics outperform other statistical measures in distinguishing stimulus-evoked responses from baseline.

## 2. Materials and Methods

The study was conducted on 30 healthy individuals (15 women, 15 men) aged between 20 and 43 years (mean = 23.63; SD = 5.6). All participants were of Caucasian ethnicity. Before the experiment, participants were informed that their involvement was voluntary, that they could withdraw at any time, and that all data would be processed anonymously. All participants provided written informed consent. In addition to consent, basic health information was collected, particularly any history of photo-epileptic seizures, because participants played the “T-Rex Rush” game during the study. The ethics committee recommended collecting these data.

EDA was recorded by measuring skin conductance using the Grove GSR Sensor V1.2. It applies a constant direct current (DC) of very low intensity (typically under 1 µA) [9] through two nickel electrodes attached to the participant’s fingers. The resulting electrodermal signal is amplified using an LM324 operational amplifier integrated within the sensor module. The sensor detects variations in skin conductance caused by sweat gland activity, reflecting sympathetic nervous system arousal. The measured signal is converted into voltage changes, subsequently digitized at a sampling frequency of 20 Hz using an Arduino Mega board’s analog-to-digital converter (ADC) with a 10-bit resolution, enabling accurate recording of subtle physiological changes [10].

Solid (gel-free) electrodes were used for EDA measurement. The same electrode set was utilized for all participants; after each session, the electrode surfaces were thoroughly washed with water and dried. This cleaning removed sweat residue and prevented cross-subject contamination, maintaining consistent electrode performance across measurements.

Electrodes were placed on the inner sides of the index and middle fingers of the participant’s non-dominant hand (on the medial phalanges). This placement allowed for comfortable movement of the other hand during the task (a computer game). Before the experiment, participants washed and dried their hands to standardize skin conditions. Participants rested the instrumented hand comfortably on a pre-prepared support pad. It is also worth noting that all subjects waited approximately 10 min in the laboratory waiting room before entering the testing area. After electrode application, an additional 5 min was allowed for EDA signal stabilization. Before starting the experiment, every participant received the exact standardized instructions, including a description of the procedure and guidelines for behavior during the study. Because a thermal imaging camera recorded the sessions, participants were also instructed to minimize bodily movements. Measurements were carried out in a room maintained at a constant temperature and under controlled environmental conditions, eliminating factors that could disrupt skin conductance (e.g., excessive sweating unrelated to stimuli). The signal was recorded using a Python 3.7 script that utilized NumPy and SciPy (for analysis and statistics) and the aseeg library (for signal filtering).

Although the Grove GSR Sensor V1.2 provided sufficient sensitivity for detecting changes associated with cognitive and emotional reactions, some limitations must be acknowledged. Compared to specialized professional-grade systems such as BIOPAC Systems or Empatica E4, the Grove GSR Sensor V1.2 offers relatively lower signal resolution and a limited signal-to-noise ratio. Professional systems like BIOPAC use advanced analog-to-digital converters (typically 16-bit or higher), ensuring more precise measurements and better detection of subtle electrodermal responses, especially useful in clinical or highly sensitive psychophysiological research contexts [11]. Similarly, wearable solutions such as Empatica E4 also provide integrated high-resolution data acquisition and advanced signal processing algorithms, enabling more robust continuous monitoring without frequent recalibrations [12]. Conversely, the Grove GSR Sensor, while affordable and accessible, can exhibit baseline drift and susceptibility to external interference, potentially necessitating more frequent calibration procedures and signal corrections, particularly in prolonged experimental sessions or scenarios demanding high precision [13]. Finally, it is worth noting that the LM324 Sensor used in the Grove GSR is not a rail-to-rail output amplifier, meaning that it cannot fully utilize the supply voltage range of its output [14]. This may limit measurement accuracy in some applications.

The procedure consisted of five consecutive stages, visually illustrated in Figure 2:Resting period 1: 1 min relaxation during which nature sounds (forest, birdsong) were played to reduce participants’ arousal levels.Task 1: Controlled breathing exercise (1 min). A pulsating green circle was displayed on-screen; participants performed deep inhalation when the circle expanded and exhalation when it contracted. The goal was maintaining calm, rhythmic breathing (low cognitive and emotional reaction task).Resting period 2: Another relaxation period (1 min) featuring nature sounds again served as a break before the next task.Task 2: Playing T-Rex Rush (approximately 5 min). Participants engaged in a simple computer game (T-Rex Rush, known from the Chrome browser), which involved controlling a running dinosaur to avoid obstacles (cacti, pterodactyls). Participants jumped using the space bar and ducked using the down arrow key. Each collision with an obstacle ended the game (a so-called “failure”), after which participants could restart, aiming for a better score. This task represented a condition of increased cognitive–emotional load, requiring concentration and evoking emotional engagement associated with avoiding failure.Resting period 3: Final calming phase (1 min), following game completion.

The study participants received standardized instructions explaining the experimental procedure. They were advised to breathe naturally, relax during nature sounds, and remain engaged during gameplay while avoiding unnecessary movements (e.g., not supporting their faces with their hands). Participants knew they could withdraw from the study at any time (participation was voluntary).

The research team selected a set of 12 statistical measures of the EDA signal potentially valuable for detecting responses to stimuli, guided mainly by publication [15], which provides a comprehensive review of statistical measures used for detecting the P300 potential in electroencephalography. Measures suitable for EDA signals were chosen from this review. The final analyzed signal features (abbreviations in parentheses) included mean (mean), standard deviation (std), variance (var), latency (lat), the highest signal value in a given time window (max), the lowest signal value in a given time window (min), latency-to-amplitude ratio (lat/max), number of zero-crossings (zero, number of sign changes after subtracting the mean), amplitude, the positive area under the curve (area+), the negative area above baseline (area−), peak-to-peak value (p-t-p), and median (med).

A specific skin conductance response (sSCR) is defined as any increase in skin conductance with an amplitude of ≥0.05 μS [16], whose onset occurs within the specified time window (the same threshold was applied in this study). When a response onset falls within this post-stimulus window and its amplitude exceeds the threshold, it is categorized as the sSCR for that stimulus. If more than one SCR waveform appears within the window, the largest (i.e., highest amplitude) is typically taken as the representative response. Conversely, if no response meeting these criteria occurs within the window, a zero is recorded (indicating the absence of a specific response to that stimulus). A sample signal measured during the experiment is shown in Figure 3. The EDA signals underwent preprocessing (high-pass filtering with a cutoff frequency of 0.05 Hz) and were divided into time segments. For further analysis, segments recorded during the T-Rex Rush game were selected, as this phase was expected to generate the most electrodermal reactions in response to stimuli. Two segments were identified (shown in Figure 4): a baseline fragment representing the resting state (4 s before stimulus onset until the stimulus) and a post-stimulus segment from 1 to 5 s after the stimulus, encompassing the expected EDA response. The choice of a 1 to 5 s window after the stimulus was deliberate, aligning with anticipated response timing. To avoid overlapping reactions, stimuli occurring within 10 s of another EDA-evoking stimulus were excluded. This prevented cumulative effects and ensured accurate baseline determination. Overall, several dozen signal segments (stimuli) from all participants were analyzed and categorized into two types generated during gameplay: participant button presses (jump or duck keys) and gameplay “failures” (collisions with obstacles resulting in the game’s termination).

The analysis was conducted on signal fragments recorded during the experiment. First, for each segment, the value of each statistical feature was calculated separately for the baseline period and the post-stimulus interval. The difference between the post-stimulus and baseline values was computed for each feature. To enable comparisons among features with different scales and units, all differences were normalized to the [0, 1] range. Normalized differences were calculated separately for each participant and feature. The resulting normalized and averaged scores represent how each statistical measure differentiates the signal, reflecting the magnitude of stimulus-induced EDA changes.

In addition to difference analysis, a separate correlation analysis was conducted to investigate relationships among signal features. Spearman’s correlation coefficient was used to evaluate correlations due to its robustness against deviations from normality. The resulting correlation matrix helped identify which features provide redundant information (highly correlated) and which measure independent signal properties. The strength of correlations was interpreted using Guilford’s guidelines [17]: negligible (|r| ⩽ 0.2), low (0.2 < |r| ⩽ 0.4), moderate (0.4 < |r| ⩽ 0.6), high (0.6 < |r| ⩽ 0.8), very high (0.8 < |r| ⩽ 0.9), and nearly perfect (|r| > 0.9).

## 3. Results

The analysis revealed significant correlations between specific pairs of EDA features, as presented in Table 1. A nearly perfect correlation (r > 0.90) was observed for the following feature pairs: peak-to-peak value and standard deviation (r = 0.98), as well as mean and median (r > 0.90). These pairs measure similar signal properties and can be used interchangeably without losing information. In contrast, the feature representing the latency-to-amplitude ratio (lat/max) showed negligible correlation with all other features. This lack of correlation suggests that this measure may capture distinct aspects of the EDA signal or might have limited usefulness as an indicator of stimulus-induced responses. The remaining feature pairs demonstrated moderate or weak correlations; for example, the mean showed a moderate positive correlation with the maximum value. Additionally, complementary features such as positive and negative areas above or below the baseline exhibited expected moderate correlations, reflecting their interdependent nature.

The averaged percentage results illustrating changes in EDA features between the baseline state and responses to stimuli are presented separately for two stimulus types: game “failures” (moments of collision) and button presses. For failures (“game overs”), the most significant observed change occurred for maximum amplitude, which increased by approximately 61.8% compared to baseline (Figure 5). Significant differences (ranging from 50% to 57%) were also noted for mean, median, and standard deviation measures. These results indicate that in response to a strong emotional stimulus (sudden failure during a task), amplitude-related signal parameters (extreme values, mean) increase more prominently than temporal or shape-related parameters. In contrast, for button presses (jump or crouch actions), the average percentage changes in EDA signal features were generally lower than those observed for failures (Figure 6). The highest increases were noted for mean, median, maximum, and minimum amplitude, with values rising between 39% and 42%. Importantly, these changes were noticeably more minor than those observed during “failures”. Features such as variance, standard deviation, latency-to-amplitude ratio, and peak-to-peak value exhibited intermediate changes (approximately 20–33%). The most minor increases were observed in latency and shape-related parameters, such as positive and negative areas and zero-crossings (7–12%).

In summary, for both intensely emotional stimuli (“failures”) and simple actions (button presses), four features best differentiated the signal: mean, median, maximum, and minimum values. Their post-stimulus values deviated significantly from the baseline (showing the highest differences). In contrast, the weakest indicators were latency, the number of zero-crossings, and the positive and negative areas, which produced the most minor changes in both conditions (nearly zero for low-emotion stimuli and approximately 30% for stronger ones). The remaining features (variance, standard deviation, latency-to-amplitude ratio, and peak-to-peak value) showed moderate differentiation, with an interesting pattern of greater effectiveness for more potent stimuli (e.g., the difference in the latency/amplitude ratio reached 56% for “failures” but only 20% for button presses).

## 4. Discussion

The results indicate which statistical EDA features are most and least helpful in detecting changes associated with cognitive load or emotional response. Mean, median, maximum, and minimum values proved to be the most sensitive to stimulus onset; these features consistently exhibited the highest differences between the baseline state and the response, regardless of stimulus type. For example, Refs. [18,19,20,21] showed that measures of overall magnitude, such as mean or median conductance, and peak SCR amplitude were the strongest predictors of high self-reported mental effort. Accordingly, these features can be recommended as the primary indicators for detecting EDA changes. Moreover, the high redundancy between mean and median (r = 0.97) suggests that, in practice, including only one of them is sufficient. Similarly, the robust correlation between the peak-to-peak value and standard deviation (r = 0.98) indicates that these two measures provide nearly overlapping information. Variance and standard deviation are also closely related (as expected mathematically), so in practice, one of them may represent both. Thus, EDA analyses can be streamlined by selecting only one feature from each highly correlated pair, simplifying modeling without significant loss of information.

The roles of latency and the response area are also noteworthy. The low effectiveness of latency (the delay in the response) as an indicator of change suggests that the time of EDA onset did not differ significantly between pre- and post-stimulus states in this task. In other words, regardless of stimulus intensity, responses occurred roughly simultaneously; hence, latency does not reflect the magnitude of arousal but rather a physiological property of the reflex. The assertion that latency does not reflect the magnitude of arousal but rather a property of the reflex is partly supported by its characterization as a physiological delay; however, the sources [6,22] do not directly compare latency with other metrics (such as amplitude) in terms of what they represent. Similarly, the number of zero-crossings and the positive/negative areas of the signal were insensitive measures, probably because they capture subtler aspects of signal shape, which in our conditions proved less significant than amplitude changes. In other experimental configurations (e.g., with more extended responses or different types of EDA signals), their utility might be more important; however, in the context of brief skin reactions to simple stimuli, they were not very informative.

The comparison of the two stimulus types provides additional insights. More potent emotional stimuli associated with game failures (“failures”) produced a more pronounced EDA increase than routine actions (button presses). This is evidenced by the highest recorded difference of 61.8% (maximum) for failures compared to 41.6% (mean) for button presses. This finding can mean that a negative emotional stimulus (an unexpected failure) engages cognitive processes (attention shifting, potential frustration) and emotional processes, eliciting a stronger physiological response. In contrast, a button press—though indicative of a decision and some cognitive load (motor planning)—does not carry a high emotional burden, resulting in a weaker skin response. This outcome is consistent with the literature: EDA responses tend to be more significant when a task is accompanied by higher emotional or stress-related arousal. EDA amplitudes are markedly larger for intense stimuli such as actual pain than for their anticipation, showing that high-intensity events provoke stronger autonomic responses [23,24]. Similarly, an unexpected game failure—perceived as negative or stressful—elicits a bigger EDA increase than a simple button press, reflecting both emotional arousal and cognitive effort [25,26]. Notably, Ref. [27] found that even watching a replay of a character’s fall produced measurable SCRs. Thus, high-arousal or aversive events consistently generate larger SCRs than neutral actions.

It is also worth noting that certain features, such as the latency-to-amplitude ratio or variance, differentiated the signal more effectively under emotional stimulus conditions (failures) than under neutral ones. For example, the latency/amplitude ratio change reached 56% for failures, compared to only 20% for button presses. This suggests that some EDA measures may be susceptible to the emotional component of a stimulus, with their diagnostic value manifesting primarily in the study of responses to stress or emotional stimuli.

Identifying four key features (mean, median, max, and min) as the best indicators of EDA response may facilitate planning future cognitive and emotional reactions studies. Researchers can focus on these measures in EDA analysis, increasing the likelihood of detecting subtle changes. In contrast, features that showed weak differentiation (latency, zero-crossings, and areas) may be omitted or treated as secondary, reducing complexity when analyzing many variables. The results of our analysis could be applied to the refinement of experimental protocols. For example, in designing studies, EDA might be used in place of questionnaires with the understanding of which signal parameters best indicate observer discomfort. In future research, employing stimuli with a more substantial emotional load (e.g., realistic, unsettling character animations) may be worthwhile to evoke more pronounced EDA responses. Our findings suggest that more potent stimuli yield more precise signals, facilitating their analysis. Additionally, by following the model of the present experiment, participants can be motivated to engage more fully (e.g., through incorporating an element of competition or symbolic rewards in the game), thereby increasing the likelihood of an emotional response and producing more discernible EDA data.

It is also important to consider external conditions affecting EDA measurement. As noted in the literature, ambient and body temperature influence the reactivity of sweat glands. Therefore, studies utilizing EDA should not be conducted in icy conditions, as skin cooling reduces response amplitude and may mask differences. Future experiments should maintain constant, comfortable environmental conditions and control factors that may affect the signal (humidity, participants’ hydration status, etc.). Empirical studies show that extreme temperatures distort EDA signals. For example, Ref. [28] reports that “extreme temperatures lead to a higher number of non-specific SCRs” in the electrodermal trace. In ambulatory measurements, Ref. [29] similarly found that higher ambient temperature increased the number of non-specific fluctuations in skin conductance.

This study’s findings are tempered by several key limitations. First, the small, demographically homogeneous sample of 30 healthy Caucasian adults limits the generalizability of our results to more diverse populations. Second, the experimental paradigm—comprising only a paced breathing protocol and a simple video game challenge—captures a narrow slice of possible cognitive and emotional loads, so the sensitivity of the highlighted EDA features may not extend to other contexts. Third, strong correlations among specific metrics (e.g., mean vs. median, peak-to-peak vs. standard deviation) suggest opportunities to streamline analyses and underscore the dependence of our conclusions on the particular feature set and analytic approach employed. Finally, because all measurements were taken under strictly controlled laboratory conditions, these results cannot be readily extrapolated to real-world or ambulatory settings, where varying environmental and physiological factors may markedly influence skin conductance readings.

## 5. Conclusions

The study aimed to determine which statistical measures of the EDA signal most effectively differentiate the signal, i.e., reveal the difference between the resting state and the response to a stimulus. This aim was achieved. The experiment involved tasks with varied cognitive and emotional reactions and provided EDA data to analyze twelve selected signal features. It was demonstrated that four of these features—mean, median, maximum, and minimum values—effectively differentiate the signal regardless of the type of stimulus. The highest levels of post-stimulus EDA changes were observed for all these measures. Additionally, it was found that there are nearly perfect correlations between pairs of features, such as mean–median and peak-to-peak value–standard deviation. This indicates redundancy; one measure from each highly correlated pair can be used instead of both, simplifying the analysis without losing significant information.

In conclusion, the presented results may be applied in further research on cognitive and emotional reactions. They enable the informed selection of EDA indicators appropriate to the experimental objectives. Conversely, if the study concerns specific aspects (e.g., the dynamics of the response), less sensitive temporal parameters might also be considered. However, their limited effectiveness must be noted. These findings are beneficial when designing experiments using EDA as an objective indicator of discomfort or cognitive engagement.

## Figures and Tables

**Figure 2 sensors-25-03300-f002:**
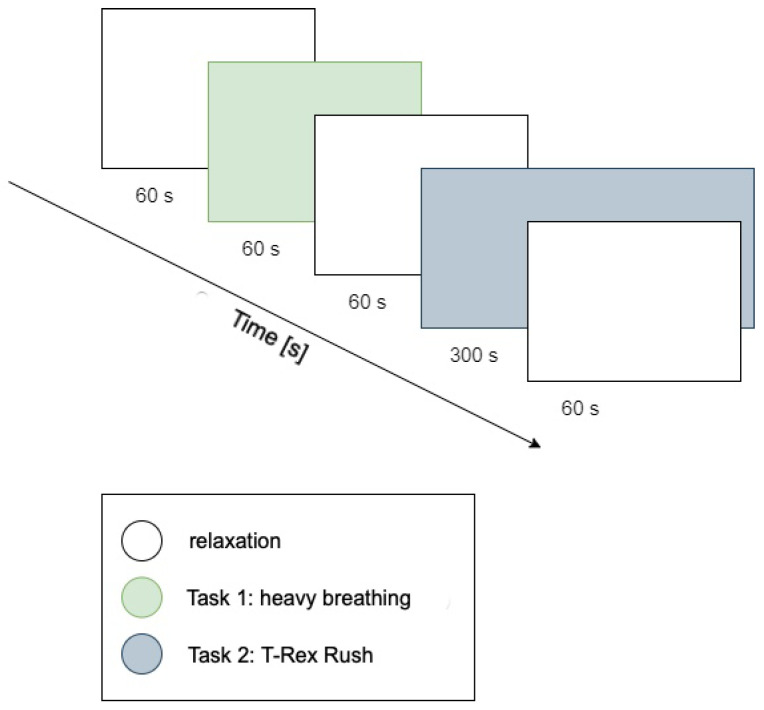
Diagram illustrating the occurrence of individual stages of the experiment over time.

**Figure 3 sensors-25-03300-f003:**
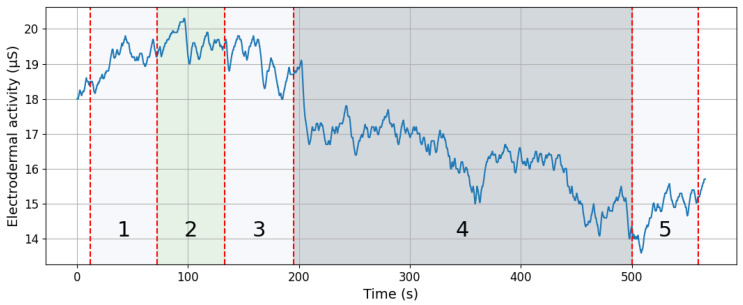
A representative electrodermal activity signal measured throughout the entire experiment. Labels 1–5 indicate all parts of the experiment described above and are also presented in Figure 2.

**Figure 4 sensors-25-03300-f004:**
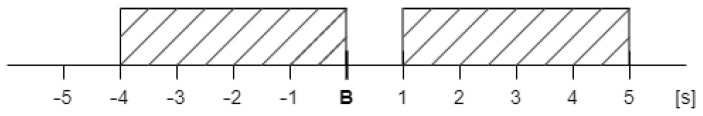
Diagram illustrating signal fragments considered in the analysis: the letter B indicates the moment of stimulus onset.

**Figure 5 sensors-25-03300-f005:**
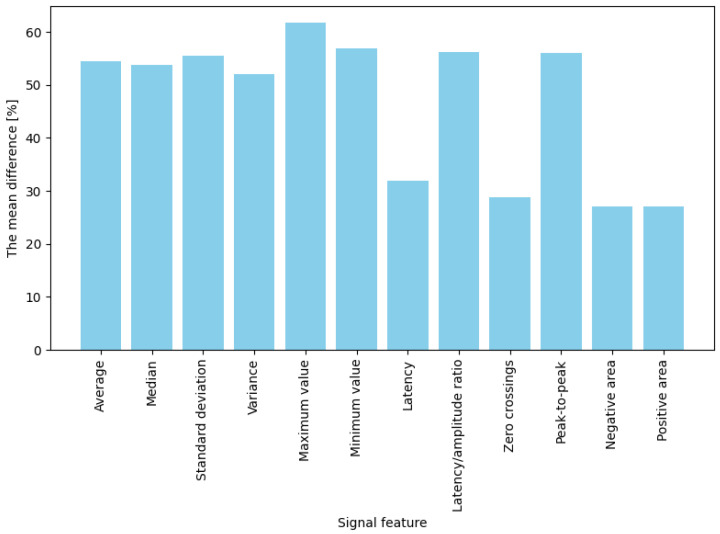
Bar plot showing the averaged differences in signal feature values before and after a “failure”, the differences are expressed as percentages.

**Figure 6 sensors-25-03300-f006:**
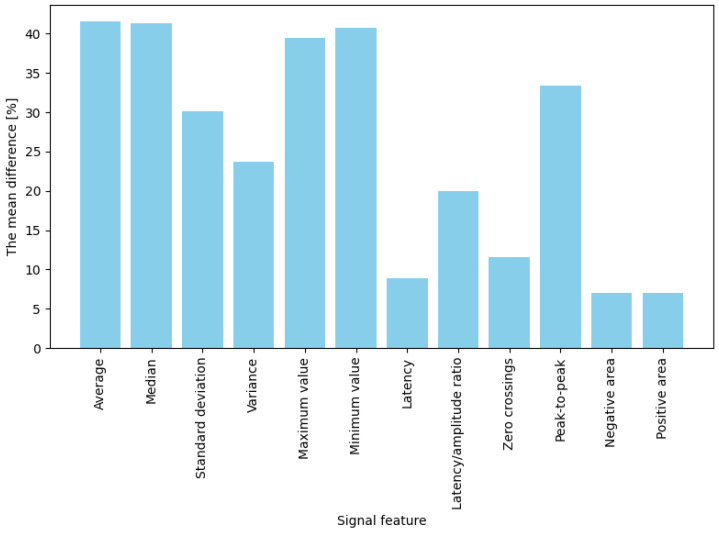
Bar plot showing the averaged differences in signal feature values before and after a button press. The differences are expressed as percentages.

**Table 1 sensors-25-03300-t001:** Correlation matrix presenting relationships between signal feature values for all participants.

	**mean**	**std**	**var**	**lat**	**max**	**min**	**lat/max**	**zero**	**area+**	**area−**	**p-t-p**	**med**
**mean**	1.00	−0.19	−0.18	0.10	0.62	0.73	−0.00	−0.01	0.67	−0.67	−0.19	0.97
**std**	−0.19	1.00	0.91	−0.13	0.61	−0.76	0.02	−0.15	−0.10	0.09	0.98	−0.17
**var**	−0.18	0.91	1.00	−0.14	0.52	−0.68	0.01	−0.10	−0.08	0.06	0.86	−0.15
**lat**	0.10	−0.13	−0.14	1.00	−0.03	0.10	−0.03	−0.05	0.09	−0.08	−0.10	0.12
**max**	0.62	0.61	0.52	−0.03	1.00	0.02	0.02	−0.13	0.43	−0.45	0.62	0.56
**min**	0.73	−0.76	−0.68	0.10	0.02	1.00	−0.01	0.10	0.47	−0.47	−0.78	0.66
**lat/max**	−0.00	0.02	0.01	−0.03	0.02	−0.01	1.00	0.02	0.03	−0.03	0.02	−0.00
**zero**	−0.01	−0.15	−0.10	−0.05	−0.13	0.10	0.02	1.00	−0.02	0.06	−0.16	−0.01
**area+**	0.67	−0.10	−0.08	0.09	0.43	0.47	0.03	−0.02	1.00	−0.81	−0.10	0.65
**area−**	−0.67	0.09	0.06	−0.08	−0.45	−0.47	−0.03	0.06	−0.81	1.00	0.08	−0.65
**p-t-p**	−0.19	0.98	0.86	−0.10	0.62	−0.78	0.02	−0.16	−0.10	0.08	1.00	−0.16
**med**	0.97	−0.17	−0.15	0.12	0.56	0.66	−0.00	−0.01	0.65	−0.65	−0.16	1.00

## Data Availability

The datasets presented in this article are not readily available because the data are currently being used in other parallel studies. Requests to access the datasets should be directed to Marcin Jukiewicz at marcin.jukiewicz@amu.edu.pl.

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
