# Peer review of "Analysis of Electrodermal Signal Features as Indicators of Cognitive and Emotional Reactions—Comparison of the Effectiveness of Selected Statistical Measures"

_sensors, 2025, doi:10.3390/s25113300_

Round 1
Reviewer 1 Report
Comments and Suggestions for Authors
Comments and Suggestions for Authors
In this study authors examined the effectiveness of some features of EDA waveforms as biomarkers for cognitive load.
The topic is interesting, the paper is well written, and the results are clearly presented. However, I have some comments, and hopefully, these could be helpful for authors and possible readers of this manuscript:
Abstract
- In the abstract, authors should indicate the rational/research question for conducting the study. Since the research question of this study is not entirely clear in the abstract.
Introduction
- Electrodermal activity is abbreviated as EDA, why are full words repeated again in this section?
- A succinct issue statement and hypothesis at the end of the introduction would strengthen the work.
Materials and methods
- How was the sample size (30) determined?
- Participants' race information is missing here, as it has been shown that African American participants show fewer EDA responses compared to others (see this article https://doi.org/10.1111/psyp.12909 for example).
- Ethics approval of research should be obtained. You should provide the decision number of the Ethics Committee of your university and the date of the decision.
- What kind of electrodes were used for (EDA) measurements, since the type of electrode is very important for EDA recordings? EDA recordings change dramatically when the electrode or the skin itself is wet. So, please mention whether the used electrodes were solid or wet jail. What about utilized electrodes? did the electrode change after each subject (new electrode for every new subject), or was one set of electrodes used for all test subjects?
- Speaking or talking is another factor that strongly elicits EDA signals. So, participants should not talk at all during recording EDA. If they talked for any reason during recording EDA, then the recorded EDA would include some additional responses due to talking, which are not associated with tasks.
- What about informed consent? Did the authors obtain a written informed consent from the participants? This should be mentioned.
- Movement is the most important artifact in EDA recordings, where it is very sensitive to the movement, either the electrodes' movement or the movement of the participant's fingers (hands). How did the authors ensure that participants avoided any body movement during EDA recordings?
- Can authors mention how specific EDA responses were obtained, as during the recording of EDA from participants, there are always non-specific EDA responses due to non-specific stimuli?
- The time between electrode placement and the start of recording EDA is of importance, as it can take time for the water content of the stratum corneum to match that of the electrode and to stabilize. Did the authors waited for some time for electrode stabilization before recording EDA? How much time was needed for the electrodes to be stabilized before EDA measurements started?
- The authors only mentioned EDA recording without mentioning which EDA parameter was recorded. There are several EDA parameters, such as the skin conductance, skin potential, and skin susceptance, see for example https://doi.org/10.1111/srt.12397
Results
- Please add a figure of a true recording of an EDA waveform.
- What was the unit of recorded EDA signals?
- The authors did not show any results of the EDA measurements.
- There are two other features (scores) of EDA responses that are missing in the analysis of this study: rise time and recovery times of the EDA responses. These indices are mentioned in Boucsein, Electrodermal activity, 2012). Why did the authors not include them in the analysis, as they are well-documented in scientific publications?
Discussion
- In general, authors did not support their discussion and the results with the already published study. Even they did not cite a single relevant study? There are huge published studies that authors can use to support their findings and discussion.
- The authors reported that “This outcome is consistent with the literature: EDA responses tend to be more significant when a task is accompanied by higher emotional or stress-related arousal.” (lines 228-230, page 9), please cite the literature.
- Authors write that “It is also important to consider external conditions affecting EDA measurement. As noted in the literature, ambient and body temperature influence the reactivity of sweat glands.” (lines 252-254, page 9). Here, the authors should cite that literature.
- What were the limitations of the study? The authors should mention if there were any study limitations in this section.
Conclusions
- The conclusion section is rather long. Some points of results and discussion are repeated, which are not necessary. Please shorten it to some main concluding points of the study.
Minor comments
- electrodermal activity>> EDA (line 9, page 1), as it is already abbreviated.
- Keywords should be alphabetically listed.
- electrodermal activity (EDA)>> EDA (line 22, page 2), as it is already abbreviated.
- electrodermal activity >> EDA (line 25, page 2).
- electrodermal activity >> EDA (line 26, page 2).
- electrodermal activity (EDA) >> EDA (lines 43-44, page 2).
- electrodermal activity >> EDA (line 281, page 9).
.
Comments on the Quality of English Language
The manuscript is well written.
Reviewer 2 Report
Comments and Suggestions for Authors
This article is interesting in its comparative analysis of a set of statistical EDA indices, but in my opinion it needs 2 revisions. The core content of the article is the analysis of the correlation between the different EDA indexes and their capabilities to differentiate the reaction in different types of events: this all remains interesting and worth publishing.
Anyway by my side the article requires 2 revisions
1) The main one.
I found that the citation of EDA as a physiological signal capable of measuring cognitive load is overstated.
- In some sections of the article, EDA is more appropriately mentioned as related to both emotional and cognitive dynamics, but never. Especially in the abstract, EDA is mentioned from the beginning as an index of cognitive load. "The article analyses the effectiveness of selected electrodermal activity (EDA) signal features as indicators of cognitive load." The same "shortcut" is used many times in the article, although sometimes, as in the last of the conclusions, the authors write (more appropriately) "In conclusion, electrodermal activity is a valuable source of information on cognitive load and emotions".
The authors cite references in the bibliography about the capabilities of EDA to detect workload, but in these referenced articles there is not sufficient evidence to support this sentence "EDA index of cognitive load" as always valid. EDA, as the authors sometimes, in some sections of the article, correctly write, is activated by emotional and/or cognitive load, it depends on specific task and conditions.For example: In the experimental task analysed, the 2 events analysed "Failure" and "Push the bottom" the EDA indexes have different mean levels on some indexes yes, but the 2 events cannot be considered one exclusively emotional and the second exclusively cognitive.
My suggestion is then to revise the text and all the definition of EDA as an index of cognitive and emotional reaction (both) ever in the whole text.
2) The little one
It seems to me that there is an exchange of the graphs shown in Fig. 4 and Fig. 5. Could authors check.
Round 2
Reviewer 1 Report
Comments and Suggestions for Authors
I have not seen the response to this comment (The authors should provide the decision number of the Ethics Committee of your university and the date of the decision). Although the authors said that "Information on the decision number issued by the ethics committee has been added.", they did not show any decision number; they only wrote "The ethics committee recommended collecting these data."
Author Response
Dear Reviewer,
Thank you for the additional corrections.
I have not seen the response to this comment (The authors should provide the decision number of the Ethics Committee of your university and the date of the decision). Although the authors said that "Information on the decision number issued by the ethics committee has been added.", they did not show any decision number; they only wrote "The ethics committee recommended collecting these data."
There was a small misunderstanding here. At the end of the article (between the Conclusions and the References), there is a section titled Institutional Review Board Statement, and that is where we included this information.
Reviewer 2 Report
Comments and Suggestions for Authors
The maior issue adressed in my previous comment has been completly adressed
Minor minor
- line 30 see a typo in twice "cognitive" repetition
- Fig.5 The graph is for "failures": then the description below has to be revised in coherence with the referred graph
- Fig. 6 It is for "button pressures"...the same as before
Author Response
Dear Reviewer,
Thank you for the additional corrections, and we apologize for not properly implementing the ones related to figure captions. We have now addressed all three comments and marked the changes in red within the text.
line 30 see a typo in twice "cognitive" repetition
Corrected and marked in red in the text.
Fig.5 The graph is for "failures": then the description below has to be revised in coherence with the referred graph
Corrected and marked in red in the text.
Fig. 6 It is for "button pressures"...the same as before
Corrected and marked in red in the text.